# Predictive and Prognostic Value of Microsatellite Instability in Gynecologic Cancer (Endometrial and Ovarian)

**DOI:** 10.3390/cancers13102434

**Published:** 2021-05-18

**Authors:** Camille Evrard, Jérôme Alexandre

**Affiliations:** 1Service d’Oncologie Médicale, CHU de Poitiers, 86021 Poitiers, France; 2Service d’Oncologie Médicale, AP-HP, CARPEM, Cochin-Hospital, Université de Paris, 75014 Paris, France; jerome.alexandre@aphp.fr

**Keywords:** endometrial cancer, ovarian cancer, mismatch repair deficient, microsatellite instability

## Abstract

**Simple Summary:**

Endometrial cancers are the cancers most affected by microsatellite instability. This phenotype confers a demonstrated sensitivity to immunotherapy and in this sense is a major parameter to know in order to manage patients. Molecular biology, therefore, has an essential role in the better knowledge of these endometrial tumors. Moreover, the microsatellite instability phenotype is very poorly understood in ovarian cancer, yet it does exist. We therefore present here a review of the literature concerning microsatellite instability in gynecological cancers (endometrium and ovaries): its diagnosis, its clinical characteristics, and its therapeutic and prognostic impact.

**Abstract:**

For endometrial cancer, a new classification is now available from ESMO, ESGO, and ESTRO based on clinical and molecular characteristics to determine adjuvant therapy. The contribution of molecular biology is major for this pathology mainly by the intermediary of deficient mismatch repair/microsatellite instability. Detection techniques for this phenotype have many peculiarities in gynecologic cancers (endometrial and ovarian) because it has been initially validated in colorectal cancer only. Endometrial cancer is the most common tumor with deficient mismatch repair, which is an important prognostic factor and a predictor of the benefit of adjuvant treatments. Concerning advanced stages, this phenotype is a theragnostic marker for using immunotherapy. Among ovarian cancer, microsatellite instability is less described in literature but exists, particularly in endometrioid type ovarian cancer. This review aims to provide an overview of the publications concerning deficient mismatch repair/microsatellite instability in endometrial and ovarian cancers, detection techniques, and clinical implications of these molecular characteristics.

## 1. Introduction

Microsatellites are small DNA sequences, coding or non-coding, with many repetitive single nucleotide or di-, tri-, or tetra-nucleotides; they are also called “short tandem repeats”. Because of this kind of repetitive structure, it is very often a carrier of replication error especially in the case of deficiency of the MisMatch Repair system, also called dMMR. These microsatellite instabilities (MSI), with an accumulation of errors, allowed to highlight the MMR dysfunction.

When cancer carries this kind of phenotype, this indicates microsatellite instable (MSI) or MMR deficient (dMMR) as opposed to microsatellite stable (MSS) and MMR-proficient (pMMR).

Tumor dMMR phenotype is related to the loss of one (or more) of the four main proteins involved in MMR process: MLH1, MSH2, MSH6, and PMS2. In most cases, this primary cause is the hypermethylation of *MLH1* gene promotor in the tumor leading to its loss of expression.

A less common cause is the inherited transmission of a mutated allele of one of the MMR genes. It is responsible for the Hereditary Non-Polyposis Colorectal Cancer (HNPCC) or Lynch syndrome that is associated with a high risk of colon cancer but also of endometrial and ovarian cancer.

Endometrial cancer is the neoplasm most commonly associated with phenotype dMMR [1]. In recent years, this phenotype has emerged as an important prognostic factor but also as a predictor of the benefit of adjuvant treatments. In advanced stages, it is a theragnostic marker for the use of immunotherapy.

## 2. Clinical Classifications of Endometrial and Ovarian Cancers

### 2.1. Endometrial Cancer

Most endometrial cancers (EC) are diagnosed at early stage (80% in stage I), with 5-year event-free and survival rates of 80 and 95%, respectively. However, the 5-year survival dramatically decreases in locally advanced (Federation of Gynecology and Obstetrics [FIGO] stages III–IVA) or metastatic (FIGO stage IVB) diseases (68% and 17% respectively) [2].

Endometrial carcinoma can broadly be divided into two types according to histology: endometrioid, affecting approximately 80% of patients, and non-endometrioid, associated with poorer prognosis. This latter type includes serous carcinoma, the most common, clear-cell carcinoma, and carcinosarcoma.

In non-metastatic stages (I–III with complete surgery), the indications of adjuvant treatment (external radiotherapy, brachytherapy, chemotherapy) were until very recently based only on clinical prognostic factors according to the recommendations of the ESMO (European Society for Medical Oncology) (Table 1, first column). These factors are the FIGO stage; the histological type (endometrioid versus non-endometrioid); and the grade and the presence or absence of vascular or lymphatic tumor emboli [2]. 

As outlined below, the molecular classification including MMR status may improve the relapse prediction and has been recently added to guidelines for post-surgical treatment in early stages (Table 1, second column) [3]. 

### 2.2. Ovarian Cancer

Among ovarian carcinoma, there are five histological sub-types: high-grade serous, low-grade serous, endometrial, clear cell, and mucinous carcinomas.

High-grade serous carcinoma (HGSC) is the most common (70% of cases), it is consistently associated with a *TP53* mutation.

Around half of these tumors harbor defects in homologous recombination (HRD) due to BRCA 1–2 defect (25% of HGSC) or other mechanisms partially unknown [4]. These HRD tumors are especially sensitive to platinum and to the poly ADP ribose polymerase (PARP) inhibitors who have revolutionized the prognosis of patients. PARP inhibitors in maintenance as a first or second line of treatment have allowed longer progression-free survival (PFS) after platinum-based chemotherapy and surgery [4].

## 3. Detection of dMMR/MSI Gynecologic Tumors

### 3.1. Detection Techniques

There are two types of techniques to detect the dMMR phenotype in daily practice: immunohistochemistry (IHC) or polymerase chain reaction (PCR) assay; these two techniques have been mainly developed in colorectal cancer (CRC).

The first technique, IHC, is based on the protein expression on tissue samples. There are four proteins in MMR system: MLH1, MSH2, MSH6, and PMS2 that assemble two by two to spot and repair DNA replication errors [5]. The MutLα complex is formed with MLH1 and PMS2, and the MutSa complex is formed with MSH2 and MSH6. First the MutSa locates a single base pair mismatch and forms a sleeve around the DNA at the site of the error and then allows the attachment of the second complex MutLα. This repair step involves many other proteins and enzymes, including DNA polymerase, which allows the excision of the defective DNA base and then the re-synthesis of DNA. The loss of expression of one of the four proteins of the system MMR reflects its loss of function, resulting in poor DNA replication with many “uncorrected” errors, which leads to the formation of ultra-muted cancers called dMMR/MSI.

If the four proteins of MMR are staining with IHC in the nucleus of tumor cells, the tumor is pMMR, and if at least one or more proteins is lost, the tumor is called dMMR (Figure 1).

The second technique is PCR assay; the purpose is the analysis of microsatellite loci according to recommendations in CRC with mono- or di-nucletotides repeats [6,7].

There are two referenced panels for PCR assay, Bethesda panel with two mononucleotide loci (Big Adenine Tract BAT-25 and BAT-26) and three dinucleotide loci (D2S123, D5S346, and D17S250). As this technique is less sensitive, its interpretation requires comparison with healthy tissue. A tumor is considered an MSI if there is instability on at least two out of five loci, if no loci is unstable, the tumor is said to be MSS.

The pentaplex panel is composed of five mononucleotide repeats quasi monomorphic: *BAT-25*, *BAT-26*, *NR-21*, *NR-24*, and *NR-27* (*MONO-27*). Its interpretation is easier because of several very homogeneous repetitions: with three markers out of five or more being instable, the tumor is MSI, and in the case of no instable marker or one, the tumor is MSS. With the pentaplex, panel healthy tissue is recommended to be used when two markers are unstable. Thanks to its characteristics, this panel is the new standard in international recommendations for MSI testing in CRC [8].

These two panels are based on capillary electrophoresis with intervention of Taq polymerase carrying out an amplification leading to a result in the form of multiple peaks, the markers being distributed according to their sizes (Figure 2). The higher peak is the reference size for one marker and is compared to its size in the general population: If the size of the peak significantly differs from the reference, it has instability [6]. All these recommendations are built for CRC, that is why for non-colorectal cancers (and gynecologic cancers) it is essential to compare tumor tissue and non-tumor tissue for MSI testing. In this case, a tumor is called “MSI” if there are two instable markers out of five [9].

### 3.2. Specificity of Detection Techniques in Endometrial Cancers and Discrepancies

The detection of microsatellite instability has mainly been studied in CRC because of a higher incidence of this type of cancer compared to gynecological cancers (endometrial and ovarian). In 2018, in the world there were more than: 1.8 million, 380,000 and 290,000 new cases of colorectal cancers, endometrial cancers and ovarian cancers, respectively [11].

The dMMR tumor rate is higher in EC compared to CRC as found in Wang’s publication in 2017, with analysis of 91 EC and 311 CRC [12]. For PCR analysis, they found 22% and 14.8% of MSI respectively for EC and CRC. Regardless of the type of tumor (EC or CRC), instability was found most of the time in all five mononucleotide loci (up to 85%), but being more often MONO27 instable in ECs and BAT26 instable in CRCs [12]. 

The main difficulty for detection of MSI in EC is that they exhibited smaller repeat number changes than CRC: shift in the reading frame caused by the repetition of unstable microsatellites is more marked in CRC than in EC with, on average, in this study, a shift to the left of −6.3 nucleotides (nt) for CRC against −2.9 nt for EC (Figure 3) [12].

Because of this characteristic, the detection sensitivity of the different tests is lower in EC, it is therefore necessary to use tests with mononucleotide markers (pentaplex type), to compare the results with those of non-tumor DNA and use a tissue sample with at least 30% tumor cells.

Very recently, Rafonne et al. published a meta-analysis with 10 studies involving a total of 3097 patients with excellent agreement (95%) between the evaluation of the MMR phenotype in IHC (use of four antibodies) and the results of the PCR in EC [13]. An ancillary analysis of Portec 1 and 2 clinical trials shows the same results with 696 patients [14]. In most cases, when a discrepancy is observed with loss of expression of MMR proteins and MSS phenotype, methylation of the MLH1 promoter or an MSI subclone is returned. It remains to be determined whether these discordant cases have the same clinical behavior as concordant dMMR cases with MSI.

Concerning the rare reverse cases of microsatellite instability (MSI) without loss of expression of proteins in IHC (pMMR), two cases (<1%) of this type were found in this series with each time a mutation of the *POLE* gene to explain this discrepancy [14].

The first-line of use of two antibodies for the detection of PMS2 and MSH6 seems to have identical performance to the use of four antibodies [13,14]. This attitude could be sufficient when MMR status is evaluated for prognostic purposes. However, for the detection of Lynch syndrome or theragnostic purpose, four antibodies are certainly still necessary [15,16].

In total, in endometrial cancers, the search for a deficiency in the MMR system is primarily based on IHC, which is a technique more readily available than molecular biology but also less expensive and which makes it possible to specify the affected gene. In addition, this technique does not require normal reference tissue and the response time is shorter.

The evaluation of microsatellite status by PCR remains necessary as a confirmation test in case of dMMR status by IHC or equivocal results.

Finally, in the event of loss of expression of MLH1 (generally associated with a loss of PMS2), it is necessary to test for methylation of the *MLH1* promoter.

Oncogenetic counselling must be proposed to all patients with dMMR tumors in relation to MSH2, MSH6 or PMS2 loss or MLH1 loss without hypermethylation of its promotor.

### 3.3. Prevalence of dMMR Phenotype among Gynecologic Cancers

With a prevalence between 20 to 30% in the early stage, the dMMR phenotype is found especially in endometrial cancers (Table 2) [1,16]. It is this localization that will be the subject of the main part of this review, with a smaller second part concerning the knowledge on dMMR ovarian tumors, in which prevalence of dMMR tumors is below 10%.

## 4. Characteristics of dMMR Endometrial Cancer

Data from The Cancer Genome Atlas (TCGA) identified four distinct molecular subclasses based on mutational load and somatic copy-number alterations (SCNAs): ultramuted associated with an inactivating mutation of the exonuclease domain of *POLE* (5%) and 232 × 10^−6^ mutations per Megabase (Mb) average.hypermuted dMMR (30%) with 18 × 10^−6^ mutations per Mb and most with *MLH1* promoter methylation.“serous-like” (20%) characterized by a mutation of *TP53* and a high number of alterations in the number of copies of genes (“copy number high”).“copy number low” with few mutations and a low number of alterations in the number of copy, MSS, and without mutation of *TP53* or *POLE* (45%) [18].

This classification had a prognostic impact: The tumors with the best prognosis were those mutated with *POLE* mutation, while a poorer prognosis was associated with “serous-like” tumors. MSI and “copy number low” tumors had an intermediate prognosis.

These TCGA data were the basis of the Proactive Molecular Risk Classifier for Endometrial Cancer (ProMisE) classification, which makes it possible to distinguish the four molecular groups by sequencing the exonuclease domain of *POLE* and the analysis by IHC of the expression of the proteins of MMR and TP53 [23].

### 4.1. Molecular Alterations Associated with dMMR Phenotype

About 95% of dMMR EC are sporadic, meaning that the MMR defect occurred primarily in the endometrial epithelium. The most common cause (75% of all dMMR EC) is the loss of MLH1 expression related to methylation of its promoter. In 5% of cases, a constitutional mutation in one of the genes of the MMR system is observed (Lynch syndrome) [1,24,25].

The constitutional mutations of MSH2 are the most frequently found in patients with endometrial cancer in the context of Lynch syndrome (40% of cases), followed by those of MLH1 and MSH6 (around 30% each). PMS2 mutations are much rarer [16].

### 4.2. Age of Women

Lynch syndrome-related endometrial cancers occur more in younger patients than in those without a constitutional mutation (median age: 54.3 to 62.3 years respectively) [26]. The prevalence of Lynch syndrome is 9% for cases occurring before age 50. However, in 25% of Lynch syndrome cases, endometrial cancer occurs after age 60 [24,27].

Tumors with methylation of the *MLH1* promoter occur more in older women (median age at diagnosis: 65 years) than dMMR tumors with mutation (median age: 59 years) but also than MSS tumors (median age 60 years) [24,28].

In the majority of EC associated with Lynch syndrome, it would be the first cancer of the spectrum (between 51 and 80% of cases). In 20% of cases, there is a synchronous ovarian cancer [16,29].

### 4.3. Family History

A history of cancer related to Lynch syndrome in a first-degree relative is associated with the risk of Lynch syndrome in a patient with EC. However, in over a third of cases of endometrial cancer linked to Lynch syndrome, no family history is found [24].

These observations have led to the proposal of universal screening for Lynch syndrome by IHC in endometrial cancer [30,31]. In the event of loss of MLH1, a test for methylation of the promoter should be performed before considering an oncogenetic consultation.

### 4.4. Body Mass Index

The prevalence of obesity was found to be lower in patients with a mutation in the MMR system compared to those with epigenetic loss of MLH1 or pMMR status (50% versus 67% and 70%, respectively) [26,28].

### 4.5. Tumor Characteristics

The histological type of dMMR endometrial carcinoma is most often endometrioid (over 90% of cases) [23]. If 30% of endometrioid carcinomas are dMMR, the proportion would be around 20% for clear cell carcinomas, 7% for carcinosarcomas, 44% for undifferentiated and dedifferentiated carcinomas [20,21,22], and much lesser in serous subtype (0–6%) [18] (Table 3). 

Endometrial cancers associated with Lynch syndrome are thought to be preferentially located in the lower segment of the uterus, but this does not appear to be the case for tumors with methylation of *MLH1* [16,32].

Deficient MMR tumors present specific histological characteristics: significant intra-tumor lymphocytic infiltrate, presence of an undifferentiated tumor contingent coexisting with a low grade contingent, and increased frequency of high grade (47%) and vascular tumor emboli [15,23,33,34].

Methylation of *MLH1*, but not MMR genes mutations, is associated with a higher FIGO stage than pMMR tumors [28]. In a meta-analysis, dMMR tumors more often exhibits lymph node involvement (stage IIIC) than low-copy number endometrioid tumors (unmutated TP53 and pMMR) [34].

In total, dMMR tumors are associated with poor prognostic factors (high grade, vascular tumor emboli, and high FIGO stage) and are more often classified at high risk of relapse according to the clinical classification of ESMO compared to low-copy number (34 versus 14%) [23].

### 4.6. Prognostic Value of dMMR Status

In the meta-analysis of studies evaluating the ProMisE classification, dMMR tumors were associated in univariate analysis with a poor prognosis in terms of overall survival compared to “copy number low” tumors (unmutated pMMR TP53 and POLE). However, this difference disappeared in multivariate analysis considering other prognostic factors (grade, stage, presence of emboli, degree of myometrium invasion, age) [34]. 

Thus, considering the clinical prognostic factors, dMMR tumors have a prognosis equivalent to “copy number low” tumors and have a better prognosis than mutated TP53 tumors.

### 4.7. Effectiveness of Adjuvant Treatments and MMR Status

Pelvic radiotherapy in high intermediate risk tumors

The Portec 2 study compared post-operative pelvic radiotherapy to vaginal brachytherapy in tumors with a high intermediate prognosis according to the ESMO classification (Table 2). Overall, the two arms were equivalent in terms of vaginal recurrence rate, but the pelvic recurrence rate was significantly higher in the “brachytherapy” arm. In a post-hoc analysis, the increased pelvic recurrence in this arm was found to be limited to TP53-mutated tumors or in those with numerous vascular tumor emboli vascular or overexpressing L1 cell adhesion molecule (L1CAM).

Thus, these data suggested that the indication of pelvic radiotherapy in tumors with a high intermediate prognosis may be limited to TP53-mutated tumors or with numerous vascular tumor emboli or overexpressing L1CAM. In dMMR tumors without embolus, brachytherapy would be sufficient [14].

Adjuvant chemotherapy in high-risk tumors

The Portec 3 study evaluated the contribution of adjuvant chemotherapy in high-risk tumors according to the ESMO classification in addition to pelvic radiotherapy (Table 2): There was a modest but statistically significant benefit of adjuvant chemotherapy in terms of recurrence-free survival and overall survival [35].

A post-hoc analysis was carried out to evaluate these results according to the molecular classification: The TP53-mutated tumors in IHC presented an absolute benefit of adjuvant chemotherapy of 25% in terms of recurrence-free survival, whereas there was no benefit for dMMR tumors [36].

## 5. Use of MMR Status in Current Practice in Endometrial Cancers

The so-called high-risk tumors according to the classification proposed by ESMO for which there is an indication for pelvic radiotherapy and chemotherapy are in particular defined by a “aggressive” histology that covers grade 3 endometrioid carcinomas; serous, clear cell, and undifferentiated carcinomas; as well as carcinosarcomas [2].

However, several studies have reported suboptimal inter-observer reproducibility even among pathologists who are experts in identifying these tumors, both for the definition of high grade and for the endometrioid/serous distinction [37].

In addition, high-grade endometrioid carcinomas are molecularly heterogeneous: in one study, 36.2% were classified as dMMR, 12.9% *POLE* mutated, 20.7% *TP53* mutated, and 30.2% “low copy number” (pMMR, *POLE* and *TP53* unmutated). In multivariate analysis including FIGO stage and age, *POLE* mutation and dMMR status were independent prognostic factors associated with better relapse-free survival of high grade tumors, while the presence of a p53 mutation was associated with a poor prognosis [38]. More rarely, low grade endometrioid carcinomas could also present with TP53 mutations, but their prognosis is less clearly defined [39].

Finally, the identification of “serous-like” tumors cannot be based only on IHC TP53 since some *POLE* or dMMR mutated tumors can present a *TP53* mutation that has no negative prognostic value [15].

For these multiple reasons, the International Society of Gyneco-Pathology in 2019 recommended the routine use of the ProMisE classification (POLE exonuclease domain sequencing, IHC TP 53, MMR) for high-grade endometrioid subgroup to better identify serous-like tumors with a worse prognosis and which require adjuvant treatment [15]. Conversely, “non-serous like” tumors could be reclassified as “low grade” and no longer be considered for chemotherapy or external radiotherapy if there is no extension beyond the uterus [15].

In 2021, ESGO-ESTRO-ESP jointly recommended to evaluate a molecular subgroup of all endometrial carcinoma and proposed new prognostic classification integrating clinical and molecular tumor characteristics (Table 2). Thus, all TP53 mutated tumors are considered at high risk and justify adjuvant chemotherapy (except in case of superficial tumor with no myometrium invasion), while this treatment can be avoided in dMMR stage I–II tumors [3].

However, data are still lacking to support therapeutic de-escalation for dMMR tumors with III–IVA stage or aggressive histology such as undifferentiated carcinoma or carcinosarcoma [22].

### Predictive Benefit of Response to Immunotherapy in Advanced/Metastatic Stages

In metastatic endometrial cancers, the frequency of dMMR status has been estimated to be 15–20% [40]. No study has reported the prognosis of metastatic cancers based on MMR status.

The high rate of mutations observed in dMMR tumors or mutated for *POLE* is responsible for the continued formation of neoantigens. In vivo, these are responsible for an immune reaction that helps slow tumor growth [41]. In fact, dMMR or *POLE* mutated endometrial cancers usually present with an inflammatory lymphocytic infiltrate [42,43].

It is now well established that dMMR tumors have a particular sensitivity to immune checkpoint inhibitors, including anti-program death 1 (anti-PD-1), anti-program death ligand 1 (anti-PDL1), and anti-cytotoxic T-lymphocyte-associated protein 4 (anti-CTLA4) [44,45,46,47]. Specifically, in endometrial cancer, several phase II trials carried out in patients pretreated with chemotherapy have found response rates with immunotherapy ranging from 27 to 57% while they are less than 10% for pMMR tumors [48,49].

In the largest published phase II study evaluating Pembrolizumab in 49 patients, the response rate was 57% and the PFS was 25 months. The median duration of response was not reached [47]. For comparison, in a second-line phase III study, chemotherapy with doxorubicin or weekly paclitaxel was associated with an 11% response rate and a PFS of 4 months [28] (Table 3).

As a theragnostic objective, it remains an issue to define which test should be used to determine MMR status: IHC, PCR or both. In metastatic colorectal carcinoma, a high rate of primary failure of immunotherapy has been observed in cases of tumors with discordant results between IHC and PCR [52]. However, data are lacking in EC. In the case of dMMR status by IHC, confirmation by PCR is desirable before considering immunotherapy.

## 6. Ovarian Cancer and MSI: Description and Prognostic Value

Overall, in the literature, the percentage of ovarian cancer (OC) with microsatellite instability is between 2 and 10% of cases but reaches 20% in the endometrioid subgroup (Table 4). Most often, it is related to methylation of the *MLH1* promoter [53].

The synchronous diagnosis of an endometrial cancer (endometrioid type) and an endometrioid ovarian cancer is a relatively common situation that could be associated with dMMR. Soliman and al reported 20% (12/59) of patients with loss of MSH2, MSH6, or MLH1 associated with an MSI-high tumor among 59 cases [54]. Another study of 32 of these patients found discordant results with a dMMR phenotype in 53.1% (17/32) of the endometrial tumors and in 31.3% (10/32) of ovarian tumors [55].

The dMMR status is much rarer in other histological types: 2–10% of clear cell cancers, <2% of serous carcinomas, and even more exceptional in mucinous carcinomas [53,61].

Deficient MMR status in endometrioid cancers did not appear to be associated with particular clinical features [53]. In a series of 36 cases of endometrioid carcinoma of the ovary, the four molecular subtypes described in endometrial cancer were found in similar proportions [62]. A larger study by Kramer and al. with 511 cases was able to find the four molecular subtypes of TCGA including 13.7% (70/511) of dMMR tumors [60]. As in endometrial cancer, dMMR ovarian endometrioid carcinomas were associated with a better prognosis than those with mutated *TP53*.

Among 80 patients with Lynch syndrome presenting with ovarian cancer, the most common histologic type was serous carcinoma followed by endometrioid type. The age of onset was younger than in the sporadic forms [63].

There are very few reports of immunotherapy in dMMR/MSI ovarian cancers. In the KEYNOTE-158 study, Pembrolizumab was associated with a response rate of 33% and median PFS of 2.3 months among 15 patients.

## 7. Conclusions

Beyond patients at risk of Lynch syndrome, evaluation of MMR status is becoming essential for all cases of EC. In routine clinical practice, the first step for the evaluation of MMR status is the immunohistochemistry. However, dMMR status should be confirmed by the evaluation of microsatellite instability by PCR.

In the early stage, the favorable prognosis value of dMMR is partially independent of clinical tumor characteristics and may allow therapeutic de-escalation in so-called “high risk” carcinoma. In addition, dMMR status is now recognized as the most promising therapeutic target in advanced EC, given the first results of immune checkpoint inhibitors. Clinical trials evaluating these drugs in early stage will start soon.

Available data suggest that MMR status should also be evaluated in endometrioid ovarian carcinoma from a prognosis and theragnostic perspective. 

## Figures and Tables

**Figure 1 cancers-13-02434-f001:**
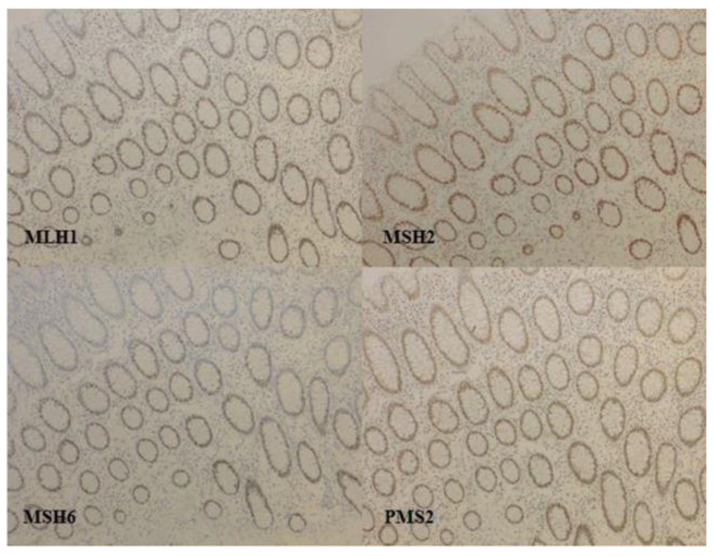
Immunochemistry of MMR proteins in colorectal cancer: a proficient MMR tumor, normal colonic mucosa, no loss of expression of MMR proteins. Antibodies used: anti-MLH1 (clone M1 Ventana^®^ Oro Valley, AZ, USA, kit Optiview^®^ for revelation, Roche Diagnostics, Meylan, France); anti-MSH2 (clone G219-1129 Ventana^®^, kit Optiview^®^ for revelation); anti-MSH6 (clone 44BD Biosciences^®^, dilution 1/500, Franklin Lakes, NJ, USA; kit Ultraview^®^ for revelation, Berkeley, CA, USA); anti-PMS2 (clone EPR 3947 Ventana^®^, ready for use; kit Optiview^®^ for revelation with amplification).

**Figure 2 cancers-13-02434-f002:**
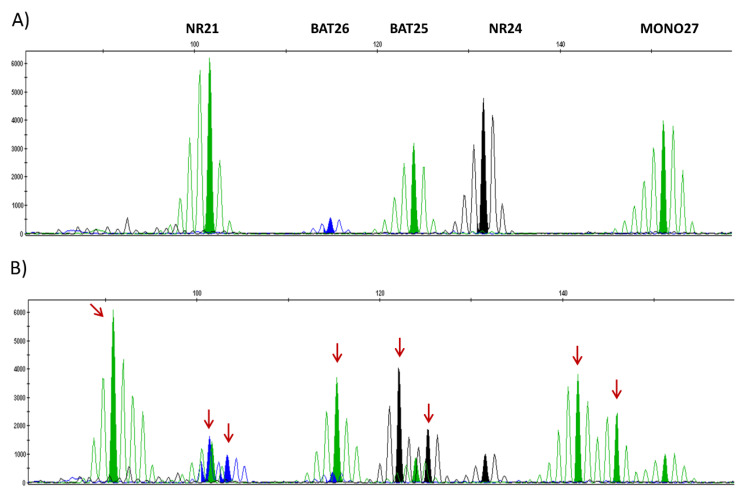
MSS/MSI profiles using the pentaplex panel [10]. Legend: (**A**): MSS profile of the five-consensus mononucleotide repeats. (**B**): MSI profile with five unstable mononucleotide repeats. Red arrows indicate unstable microsatellites.

**Figure 3 cancers-13-02434-f003:**
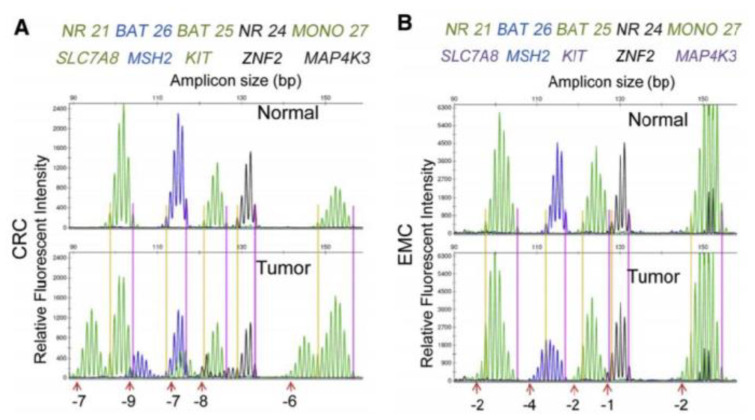
Differences in microsatellite instability (MSI) profiles between CRC and EMC [12]. Legend: MSI: microsatellite instability; CRC: colorectal cancer; EC: endometrioid cancer. (**A**): MSI profile of a representative MSI high (MSI-H) CRC with its paired normal control. Shifts in microsatellite repeat lengths are labeled at the bottom (e.g., gene *NR21/SLC7A8*, −7 nt). (**B**): MSI profile of a representative MSI-H EMC compared with its paired normal control. Shifts in microsatellite repeat lengths are labeled at the bottom (e.g., gene *NR21/SLC7A8*, −2 nt).

**Table 1 cancers-13-02434-t001:** New risk groups to guide adjuvant therapy use.

Risk Group	2016 ESMO Classification Based on Clinical Characteristics [2]	2021 ESGO ESTRO ESP Classification Based on Clinical and Molecular Characteristics [3]
Low	Stage IA endometrioid + low grade + LVSI negative	Stage I–II *POLE*mut endometrial carcinoma, no residual disease
		Stage IA dMMR/NSMP endometrioid carcinoma + grade 1–2 + LVSI negative or focal
Intermediate	Stage IB endometrioid + low-grade + LVSI negative or focal	Stage IB dMMR/NSMP endometrioid carcinoma + grade 1–2 + LVSI negative or focal
		Stage IA dMMR/NSMP endometrioid carcinoma + grade 3 + LVSI negative or focal
		Stage IA p53abn and/or non-endometrioid (serous, clear cell, undifferentiated carcinoma, carcinosarcoma, mixed) without myometrial invasion
High-intermediate	Stage I A-B endometrioid low grade + substantial LVSI,	Stage I dMMR/NSMP endometrioid carcinoma + substantial LVSI, regardless of grade and depth of invasion
	Stage IA endometrioid high-grade, regardless of LVSI status	Stage IB dMMR/NSMP endometrioid carcinoma grade 3, regardless of LVSI status
		Stage II dMMR/NSMP endometrioid carcinoma
High	Stage II–IVA with no residual disease	Stage III–IVA dMMR/NSMP endometrioid carcinoma with no residual disease
	Non-endometrioid (serous, clear cell, undifferentiated carcinoma, carcinosarcoma, mixed) with myometrial invasion, and with no residual disease	Stage I–IVA p53abn endometrial carcinoma with myometrial invasion, with no residual disease
	Stage IB endometrioid G3	Stage I–IVA NSMP/dMMR serous, undifferentiated carcinoma, carcinosarcoma with myometrial invasion, with no residual disease
Advanced Metastatic	Stage III–IVA with residual disease	Stage III–IVA with residual disease of any molecular type
	Stage IVB	Stage IVB of any molecular type

Legend: ESMO, European Society of Medical Oncology; ESGO, European Society of Gynaecological Oncology; ESTRO, European Society for Radiotherapy and Oncology; ESP, European Society of Pathology; LVSI, lymphovascular space invasion; p53abn: p53 abnormal; dMMR: Mismatch Repair Deficient; NSMP: non-specific molecular profile; *POLE*mut: polymerase ε mutated.

**Table 2 cancers-13-02434-t002:** dMMR status in endometrial cancers.

Tumoral Type	% of dMMR Tumor		
	NGS [17]	PCR (MSI)	IHC
Endometrial carcinoma	32% (*n* = 542)	24% (*n* = 696) [1]	28% (*n* = 696) [1]
Endometrioid		25% (*n* = 679) [1]	
Serous	0 (*n* = 53) [18]	6% (*n* = 17) [1]	
Carcinosarcoma	3.5% (*n* = 57)		18% (*n* = 22) [19]
			7% (*n* = 231) [20]
Clear cells			19% (*n* = 32) [21]
Un- and dedifferenciated			44% (*n* = 73) [22]

Legend: dMMR, deficient MisMatch Repair; NGS, next generation sequencing, PCR, polymerase chain reaction; IHC, immunohistochemistry.

**Table 3 cancers-13-02434-t003:** Main trials evaluating immunotherapy in MSI endometrial cancer.

Trials	Line of Treatment	Evaluated Treatments	Population	Number of Patients	Objective Response Rate (%) (95% CI)	Duration of Response (Months)	PFS (Months)	OS (Months)
Marabelle et al., 2020 [47]	≥2	Pembrolizumab	dMMR	49	57.1% (42.2 to 71.2)	NR (2.9 to 27.0+)	25.7 (4.9 to NR)	NR (27.2 to NR)
Oaknin et al., 2020 [50]	≥2	Dostarlimab	dMMR	179	44.7% (34.9–54.8)	NR (2.6–28.9)	/	NR
			pMMR	161	13.4% (8.3–20.1)	NR (1.5–30.4)	/	NR
Antill et al., 2019 [51]	≥1	Durvalumab	dMMR	35	40% (26–56)	/	/	/
			pMMR	36	3% (1–14)	/	/	/

Legend: dMMR, deficient MisMatch Repair; pMMR, proficient MisMatch Repair; NR, not reached; OS, overall survival; PFS, progression-free survival.

**Table 4 cancers-13-02434-t004:** Main references concerning dMMR/MSI status in ovarian cancers.

Trial	Evaluable Patients	All Comers	Endometrioid OC
Fraune et al., 2020 [53]	478	10/478 (2.1%) (IHC)	8/35 (22.8%) (IHC)
		9/478 (1.8%) (PCR)	8/35 (22.8%) (PCR)
Xiao et al., 2018 [56] *	419	29/419 (6.9%) (IHC)	15/98 (15.3%) (IHC)
Aysal et al., 2012 [57]	71	/	7/71 (10.0%) (IHC)
			7/71 (10.0%) (PCR)
Rambau et al., 2016 [58]	612	29/612 (4.7%) (IHC)	25/181 (13.8%) (IHC)
Hollis et al., 2020 [59]	112	/	20/112 (17.5%) (NGS)
Kramer P et al., 2020 [60]	511	/	13.7% (IHC)

Legend: IHC was immunohistochemistry with Bethesda panel; PCR, polymerase chain reaction; NGS, next generation sequencing; OC, ovarian carcinoma. ***** Endometrioid and serous OC were combined.

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
