# Peer review of "Predictive and Prognostic Value of Microsatellite Instability in Gynecologic Cancer (Endometrial and Ovarian)"

_cancers, 2021, doi:10.3390/cancers13102434_

Round 1

Reviewer 1 Report

the present manuscript represents a review regarding microsatellite instability in endometrial carcinoma. This topic is quite interesting in light of the possibility of immunotherapy in recurrent disease. Despite this, the review does not represent any particular novelty relative to the recent literature.
I suggest reporting immunohistochemistry figures relative to MMR-deficient endometrial carcinoma.
I also strongly suggest citing some recently published work in MDPI reviews as well.

  • Dondi, G et al. “An Analysis of Clinical, Surgical, Pathological and Molecular Characteristics of Endometrial Cancer According to Mismatch Repair Status. A Multidisciplinary Approach.” International journal of molecular sciences vol. 21,19 7188. 29 Sep. 2020, doi:10.3390/ijms21197188
  • Siemanowski, Janna et al. “Managing Difficulties of Microsatellite Instability Testing in Endometrial Cancer-Limitations and Advantages of Four Different PCR-Based Approaches.” Cancers vol. 13,6 1268. 12 Mar. 2021, doi:10.3390/cancers13061268
  • Gallon, Richard et al. “How Should We Test for Lynch Syndrome? A Review of Current Guidelines and Future Strategies.” Cancersvol. 13,3 406. 22 Jan. 2021, doi:10.3390/cancers13030406
  • De Leo, A et al. “ARID1A and CTNNB1/β-Catenin Molecular Status Affects the Clinicopathologic Features and Prognosis of Endometrial Carcinoma: Implications for an Improved Surrogate Molecular Classification.” Cancers vol. 13,5 950. 25 Feb. 2021, doi:10.3390/cancers13050950

Author Response

We added few references proposed by reviewer :

Dondi, G et al.; Lines 248 and 268  : endometrial cancer dMMR are more often for yonger patients with lower BMI. (Ref 26)

Siemanowski, Janna et al.; Line 127 : it explains different technics of PCR to determine MSI status in endometrial cancer. (Ref 7)

Gallon, Richard et al.; Line 262 : this article talk about systematic screening for Lynch syndrome. (Ref 31)

Reviewer 2 Report

The authors give an overview of deficient mismatch repair / microsatellite instability in endometrial and ovarian cancers and their clinical implications. This review is well written and no need additional revision.

Minor point

Lane 103, 104   MutSa -> MutSa

Author Response

No modification

Don’t understand this minor point “Lane 103, 104   MutSa -> MutSa”

Reviewer 3 Report

The review is very comprehensive and informative.  What is very helpful in this work is the comparison to the MSS/MSI in CRC which is more profound.  

Some minor issues are with Figure 1- Needs to be more descriptive; i). what is the name of the tumor for the sake of future publications ii). what are the antibodies used in the IHC?

There are also a few minor areas where the English grammar needs some tweaking to help in disseminating the information:

  1. line 28- "little" should be "small"
  2. line 34- "it says" should be "indicates"
  3. line 120- "They are" should be "There are"

Author Response

Figure 1- Needs to be more descriptive; i). what is the name of the tumor for the sake of future publications ii). what are the antibodies used in the IHC?

We add description of figure in legend : Immunochemistry of MMR proteins in colorectal cancer: a proficient MMR tumor, normal colonic mucosa, no loss of expression of MMR proteins. Antibodies used :  anti-MLH1 (clone M1 Ventana®, kit Optiview® for revelation); anti-MSH2 (clone G219-1129 Ventana®, kit Optiview® for revelation); anti-MSH6 (clone 44BD Biosciences®, dilution 1/500; kit Ultraview® for revelation); anti-PMS2 (clone EPR 3947 Ventana®, ready for use; kit Optiview® for revelation with amplification)

Minor issues with English grammar are modified :

    line 28- "little" replaced by "small"

    line 34- "it says" replaced  by"indicates"

    line 120- "They are" replaced by "There are"